# Synthesis and Biological Evaluation of Potential Oncoimmunomodulator Agents

**DOI:** 10.3390/ijms24032614

**Published:** 2023-01-30

**Authors:** Raquel Gil-Edo, Sara Espejo, Eva Falomir, Miguel Carda

**Affiliations:** Inorganic and Organic Chemistry Department, University Jaume I, E-12071 Castellón, Spain

**Keywords:** PD-L1, c-Myc, IL-6, multitarget inhibitors, immunomodulation, angiogenesis, small molecules, co-cultures, flow cytometry

## Abstract

Fourteen triazole-scaffold derivatives were synthetized and biologically evaluated as potential oncoimmunomodultator agents by targeting both PD-L1 and c-Myc. First, the antiproliferative activity of these molecules on the monocultures of several tumor cell lines (HT-29, A-549, and MCF-7) and on the non-tumor cell line HEK-293 was studied. Then, the effects on the mentioned biological targets were also evaluated. Finally, the effect on cancer cell viability when the molecules were co-cultured with immune cells (Jurkat T cells or THP-1) was also determined. Compounds bearing a bromoophenyl group were selected because of their excellent results, and their effect on IL-6 secretion was also studied. In conclusion, we found compounds that are capable of downregulating c-Myc, as well as influencing and altering the distribution of PD-L1 in tumor cells; the compounds are thus capable of influencing the behavior of defensive cells towards cancer cells. *p*-Bromophenyltriazol **3** is the most active of these as a PD-L1 and c-Myc downregulator and as a potential immunomodulator agent. Moreover, it exhibits an interesting action on inflammation-related cytokine IL-6.

## 1. Introduction

Tumors are not just masses of isolated cancer cells, as they are able to interact with the surrounding cells and with the extracellular matrix, thus generating the tumor’s microenvironment (TME). In a tumor microenvironment, the extracellular matrix serves as a support on which cells can adhere and grow [1]. Furthermore, non-malignant TME cells often play a very active role in promoting carcinogenesis through a complex dynamic network of intercellular communication with cancer cells based on the secretion of cytokines, chemokines, growth factors, inflammatory factors, etc. In fact, the evolution, structure, and activities of cells in TME have many parallels with the processes of wound healing [2] and with inflammation [3]. A recognition of the importance of the TME in the progression of cancer has led to a shift from a tumor-centered view to that of a tumor as a complex ecosystem in which non-malignant cells and molecular components are as influential as malignant cells in the evolution of cancer and metastasis [4]. A feature of the TME is that alterations in just one of its components can cause a dramatic reorganization of the entire system. Prominent among these factors is the ability of cancer cells to evade attack by the immune system, to activate angiogenesis, and to promote inflammation. All these altered processes are interconnected and have synergistic effects that help the expansion of tumors [5]. In this regard, it has been observed that tumor cells overproduce several proteins, such as programmed death-ligand 1 (PD-L1), that cause immune cell dysfunction [6]. Moreover, the overproduction of proteins that activate the process of angiogenesis leads to the formation of an expanded microvascular bed that facilitates the entry of increased nutrient supply and favors the entry of inflammatory cells, thus facilitating the local production of cytokines, chemokines, and metalloproteinases in the tumor mass [2]. The secretion of pro-inflammatory factors by tumor cells, combined with the influx of inflammatory immune cells, reshapes the TME composition by blocking anti-tumor immunity [7] and reprogramming the activity of immune cells such as macrophages. These immune cells, residing within the tumor microenvironment, are termed tumor-associated macrophages (TAMs) [8]. In TAMs, proinflammatory cytokines such as IL-6 [9], which are overproduced in the TME due to their release by tumor cells, together with some defensive cells, play a crucial role in tumor growth and metastasis. Recently, evidence has been found of a beneficial role of IL-6. In this role, IL-6 opposes tumor growth by mobilizing lymphocytes and anti-tumor T cell immune responses against tumor expansion [10]. Therefore, interference with any element of the TME provides a new opportunity in the fight against cancer [11].

For the last ten years, our research has been focused on the development of potential anticancer agents for application in targeted therapies [12,13,14]. Moreover, we have recently been fully engaged in the screening of compounds that are able to simultaneously block biological targets of special relevance not only in the cancerous process, but also in the maintenance of the tumor microenvironment (TME). The target proteins we are mainly focusing on are implicated in three of the ten characteristics common to all cancer cells [15,16]: evasion and reprogramming of the immune system [17,18], the activation of neovascularization or angiogenesis [19], and the promotion of inflammation processes. These proteins include PD-L1, which is related to the immunosuppressive capacity of tumor cells; VEGFR-2 (vascular endothelial growth factor receptor), which plays a key role in the formation and growth of new tumor blood vessels; and c-Myc protein, which intervenes in numerous processes that are altered in tumor cells, such as cell cycle, angiogenesis, or overexpression of oncoproteins [20]. In this sense, overexpression of c-Myc is considered to be responsible for the increased expression of immune control agents, thus favoring the evasion of immune surveillance [21]. Some studies have revealed that high levels of c-Myc result in the increased expression of PD-L1, thus allowing the suppression of both innate and adaptative immune responses and favoring cancer progression and metastasis [22].

Using antagonist antibodies to block PD-1/PD-L1 interaction has been a breakthrough in cancer treatment, to such an extent that it has become the gold standard of treatment for some types of cancer [23]. For example, nivolumab is used for the treatment of non-small-cell lung cancer (NSCLC) [24], prembrolizumab for head and neck squamous cell carcinoma (HNSCC) [25], and avelumab for metastatic urothelial carcinoma (UC) [26].

To date, the PD-1/PD-L1 pathway inhibitors available in clinical practice have all been mAbs. However, despite the amazing initial therapeutic response, a durable response has not yet proved to be as encouraging, and prolonged survival is not promising. In addition, monoclonal antibodies present clinical disadvantages, such as in their safety, low oral bioavailability, and difficult and expensive production, which limit their clinical use. Therefore, the development of small-molecule inhibitors is a very promising area of work, as these would allow for overcoming the above drawbacks. In this vein, small-molecule agents that modulate the PD-1/PD-L1 axis have already been described in the literature and are being evaluated in preclinical and clinical trials. For example, CA-170 is a small-molecule inhibitor of the PD-1/PD-L1 axis that is designed from a peptide derived from a region located at the interface of PD-1 and PD-L1. It was the first small-molecule inhibitor of PD-L1 to enter clinical trials in 2016 for the treatment of lymphoma and several advanced solid tumors [27]. Another small-molecule inhibitor is INCB-086550, whose clinical trials were initiated in 2018 for the treatment of urothelial cancer and renal cancer [28].

The development of these molecules had its starting point in a study carried out by P. Holak’s group from Bristol-Meyers Squibb, which resulted in the elucidation of the crystal structures of the PD-L1 dimer complex with BMS small-molecule inhibitors (see BMS-8 structure shown in Figure 1) [29]. According to the pharmacophore model that was developed for PD-1/PD-L1 inhibitors, there are three important components: a first component that occupies the hydrophobic cavity of PD-L1 and into which a biphenyl system fits very well, a second component that occupies a central planar position and serves as a link between the other two components, and a third component that is positioned in a polar cavity and can be extended further than the previous ones [30].

However, the design of small-molecule PD-1/PD-L1 inhibitors remains a challenge, because the hydrophobic nature of the binding interface might also result in the generation of “false positive” hits during in vitro screening processes. Emerging evidence suggests that numerous underlying mechanisms are involved in the specific mode of action of small-molecule PD-L1 inhibitors. It seems that theses inhibitors have the potential to simultaneously suppress not only a key oncogenic signaling pathway, but also PD-L1 expression and/or activity, through the degradation or downregulation of membrane, extracellular, and/or cytosolic PD-L1. In this context, it is quite interesting to note that PD-L1 expression is regulated by a wide range of transcription factors, and particularly c-Myc, whose expression is often significantly correlated with that of PD-L1 [31].

The main goals of the work we present here are the development of molecules that are capable of simultaneously inhibiting PD-L1 and c-Myc [32], the study of their effect on immune cells, and in addition, the study of their effect on the secretion of IL-6 to culture media.

For these objectives, we took advantage of our previous studies involving the discovery of new multitarget anticancer entities bearing a triazole core and the pharmacophore model developed by P. Holak’s group (see Figure 1) [17,18,19,29]. These kinds of molecules (E-1 and E-2 structures in Figure 1) proved to be good or moderate inhibitors of both PD-L1 and c-Myc [17,18,19]. We carried out some docking studies, starting from the RCSB Protein Data Bank with PDB ID 5J89 for the PD-L1 system [29]. These studies revealed that triazole derivatives E-1 and E-2 (see Figure 1) fit well at the PD-L1 binding site, which has been previously described as the BMS-202 binding site. Our docking studies revealed that the aromatic A-ring and the triazole unit established π-stacking interactions with aromatic residues of PD-L1 chain B, and the aromatic B-ring fit into a hydrophobic pocket in the chain A of the targeted protein. Based on these pharmacophoric features and Holak’s pharmacophore model for PD-1/PD-L1 inhibitors, we designed new biaryl structures in which the aromatic A-ring is directly attached to the triazole unit (see E-3 structures in Figure 1). The spacer between the triazole and the aromatic B-ring is shortened by introducing an ethylene group, so that the distance between biaryl and the B phenyl group is similar to the that BMS-8, and finally, an extended chain beyond the aromatic ring B is added (see Appendix A).

## 2. Results

### 2.1. Synthetic Work

Figure 1 shows the synthetic route followed to obtain the newly designed triazole compounds **1**–**14** (see Figure 1). The synthesis of these triazole derivates began with the reduction of commercially available 1,4-phenyldiacetic acid **15**. This reaction gave rise to diol **16,** which was monoprotected upon reaction with 4,4′-dimetoxytrityl chloride to provide alcohol **17** [33]. In turn, alcohol **17** was converted into tosylate **18,** which upon reaction with sodium azide led to azide **19**. The reaction of azide **19** with alkynes **20–26** [34] allowed us to achieve hydroxyl protected triazole compounds **1**, **3**, **5**, **7**, **9**, **11** and **13**. Finally, deprotected triazole derivatives **2**, **4**, **6**, **8**, **10**, **12** and **14** were obtained by removal of the protecting group under acidic conditions.

### 2.2. Biological Evaluation

#### 2.2.1. Study of the Effect on Cell Viability

The effect of triazole derivatives **1**–**14** on the cell viability was investigated by MTT assay, and IC_50_ values were determined towards the human tumor cell lines HT-29 (colon adenocarcinoma), A-549 (pulmonary adenocarcinoma), and MCF-7 (breast adenocarcinoma), as well as towards the non-tumor cell line HEK-293 (human embryonic kidney cells). Neither of the compounds were active, and we found that all of them exhibited IC_50_ values above 300 µM in all tested cell lines.

#### 2.2.2. Study of the Effect on the Expression of Total PD-L1 and c-Myc in Tumor Cells

The effect of these triazole derivates on c-Myc and PD-L1 was measured in three different cancer cell lines (HT-29, MCF-7, A-549) by flow cytometry. Thus, cells were incubated for 24 h with the corresponding compound at a concentration of 100 μM. Dimethyl sulfoxide (DMSO)-treated cells were used as the negative control, and a PD-L1 inhibitor, BMS-8 [29,31], was used as a reference control. After permeabilization and fixation, cells were treated with anti-PD-L1 conjugated to AlexaFluor^®^647 and anti-c-Myc conjugated to fluorescein to determine the amount of both targets related to the negative control. Table 1 show the results in terms of the percentage of the detected target in relation to the negative control. This assay was carried out on cancer cell lines HT-29, MCF-7, and A-549.

As can be seen from Table 1, the expression for each compound changed depending on the cell line being studied. The HT-29 cancer cell line showed the best results. However, in general, protected bromo derivates (**3**, **5**, **7**) were the most active as dual inhibitors in the three cancer cell lines. In addition, the results clarify that compounds bearing the protecting group **1**, **3**, **5**, **7**, **9**, **11** and **13** have a greater inhibitory activity than the deprotected derivatives.

As regards the action on c-Myc, in addition to **3**, **5** and **7**, we found that *m*-MeO derivative **11** inhibited this target by half in HT-29 and in A-549 (50 and 55%, respectively). In this last cell line, A-549, methoxy derivatives **9** and **13** were also quite active, with inhibition rates of 40 and 50%, respectively. As regards the action on total PD-L1, the protected compounds were the most active in the HT-29 and in MCF-7 cell lines, although the effect in this latter line was very weak (around 10%), whereas in HT-29, the inhibitory effect reached to 50–60%. On the other hand, the unprotected compounds were the most active ones in the A-549 cell line, with inhibitions ranging from 50–60%. Even so, the *p*-Br protected **3** was as active as the unprotected **4**.

We also observed the effect of triazole derivatives **3**, **5** and **7** on both targets in MCF-7 by immunofluorescence microscopy (see Figure 2). We stained both targets with anti-PD-L1 conjugated to AlexaFluor^®^647 (red in Figure 2), as well as anti-c-Myc conjugated to fluorescein (green in Figure 2), to determine the amount of both targets. The nucleus was stained with Hoechst (blue in Figure 2).

We observed that in the DMSO control, c-Myc was located in the nucleus of the cell and the nucleus membrane nearby, and PD-L1 was preferentially positioned in the membrane or cytosol, favoring the contact between surrounding cells. In BMS-8-treated cells, PD-L1 was located preferentially in the nucleus, so that the PD-L1 membrane was altered or even disappeared, whereas c-Myc was not altered. When cells were treated with compound **3,** the same was observed, as PD-L1 accumulates in the cytosol, and c-Myc is preferentially in the nucleus of the cells. On the other hand, when cells were treated with compound **5,** we observed that PD-L1 was accumulated in the interface of cells and in the nucleus, whereas c-Myc aggregated around the nucleus membrane. Finally, when cells were treated with compound **7,** we observed that PD-L1 was preferentially in cytosol, and c-Myc aggregated in the nucleus of the cells.

We also studied the effect of compounds **3**, **5,** and **7** on both c-Myc and PD-L1 gene expression by RT-qPCR. For this study, we treated MCF-7 cells with 100 µM of the abovementioned compounds, and DMSO was used as a negative control and BMS-8 as a reference compound. After 48 h, cells were lysated, and after the extraction of their mRNA, a RT-pPCR assay was performed as described in the Experimental section. The results are shown in Table 2.

This study demonstrated that these compounds are able to downregulate both targets. As regards c-Myc gene expression, compound **3,** bearing a *p*-bromophenyl group, showed to be the most active, because it was able to inhibit around 85% of the gene expression. In addition, compound **5** was as active as the reference compound BMS-8. As regards PD-L1, compounds **3** and **7** were able to downregulate gene expression, improving BMS-8 effect.

Finally, we studied the correlation between the dose of compound **3** and the effect on the targets. We observed that in HT-29, concentrations below 50 µM had no effect on both targets, whereas in MCF-7, a non-dependence-dose effect was observed on c-Myc, and a maximum effect was seen on PD-L1 at 10 µM (see Table 3). For lower concentrations, the effect was very mild.

#### 2.2.3. Study of the Effect on Viability of Cancer Cells in Co-Culture with Jurkat T Cells

The effect of the synthesized compounds on tumor cell proliferation in the presence of PD-1-expressing Jurkat T cells was also studied. This assay was carried out, as in the previous experiments, in the HT29, MCF7, and A549 cell lines (see Table 4). Interferon γ-stimulated cancer cells were treated for 24 h with the compounds at 100 μM in presence of Jurkat T cells. Then, cells were counted using flow cytometry. The percentage of living cells relative to non-treated co-cultures (control) was established for both cancer and immune cells in every co-culture.

The effect on A-549 cell viability when co-cultured with Jurkat T cells was moderate for the unprotected derivatives **4** (*p*-Br), **6** (*m*-Br), **8** (*o*-Br), **12** (*m*-MeO), and **14** (*o*-MeO), with inhibition rates around 30%. In addition, we observed that protected and unprotected *p*-Br derivatives **3** and **4** were both active, whereas protected and unprotected *p*-OMe derivatives **9** and **10** showed no activity. The remaining compounds had no effect on the A-549 cell line. From these observations, we can assume that the more polar the molecule (free hydroxy group), the greater the effect on A-549 cell viability when they are cultured together with Jurkat T cells. On the other hand, the effect on HT-29 cell viability when co-cultured with Jurkat T cells was higher than that on A-549. In this case, the protected derivatives, **5** (*m*-Br), **7** (*o*-Br), **9** (*p*- MeO), and **11** (*m*- MeO), were more active than the unprotected ones, showing inhibition rates on cell proliferation of around 50–40%. Contrary to that which we observed for A-549, in the HT-29 cell line, both protected and deprotected *p*-Br derivatives **3** and **4** were not active, whereas *m*-MeO derivatives **11** and **12** were both active, showing around 40% inhibition rates. The most active compound was derivative **7** (*o*-Br), showing around 50% inhibition of cell viability. Finally, on the MCF-7 cell line, all the compounds showed relatively high inhibition rates when co-cultured with Jurkat T cells. In general, protected derivatives were more active than deprotected ones, with inhibition rates of around 60% and 30%, respectively. The most active derivative was protected compound **5** (*m*-Br 5), with a 75% inhibition rate, followed by *p*- and *o*-Br derivatives **3** and **7**, and both *o*-MeO **13** and **14**, which inhibited almost 60% of cancer cell viability in the presence of Jurkat T cells. Derivative **11** (*m*-MeO) was also outstanding; it was able to decrease cell viability in half.

#### 2.2.4. Study of the Effect on the Membrane PD-L1 and VEGFR-2 in Co-Culture Cancer Cells

We determined by flow cytometry the effect of these compounds on the expression of membrane PD-L1 in cancer cells when they were co-cultured with immune cells. Thus, interferon γ-stimulated cancer cells were treated for 24 h with the compounds at 100 μM in presence of Jurkat T cells. Then, cells were fixed and treated with anti-PD-L1 conjugated to AlexaFluor^®^647. We took advantage of the versatility of flow cytometry to simultaneously measure the expression of membrane VEGFR-2 by using anti-VEGFR-2 conjugated to fluorescein. Table 5 shows the relative amount of membrane PD-L1 and VEGFR-2 related to the negative control (non-treated co-cultures).

Table 5 shows that, in general, neither of the derivatives was very active against membrane VEGFR-2 in any cell line. On the other hand, the effect on membrane PD-L1 was higher in MCF-7 than in HT-29 and was very mild in A-549. As regards the effect on membrane PD-L1 in HT-29, deprotected bromo derivatives **4**, **6** and **8**, together with protected *p*-MeO **9** and both *o*-MeO derivatives **13** and **14**, exerted a higher action, showing membrane PD-L1 inhibition rates of about 40–50%. As regards the effect on PD-L1 in MCF-7, protected bromo derivatives **3**, **5** and **7**, together with methoxy derivatives **9–13**, exerted excellent inhibition rates, with values ranging from 60–70%. This could explain why MCF-7 was the most sensitive cell line to the effect of the synthetic compounds in enhancing T cells’ antiproliferative action towards this cancer cell line (see Table 4). Finally, we could assume that *p*-bromo compound **3** is the one that combines the most versatile action, since it is active in all three cancer cell lines and can inhibit membrane PD-L1 by half in MCF-7, showing inhibition rates of around 20% for the rest of the cell lines and on both targets.

#### 2.2.5. Study of the Effect of Selected Triazole on Viability of Cancer Cells in Co-Culture with THP-1

The effect of the synthesized compounds on tumor cell proliferation in the presence of human monocytic leukemia cell line THP-1 was also studied. This assay was carried out using MCF-7 as the cancer cell line and different proportions of immune cells. Standard proportions for this assay were 1:5 cancer/immune, and we also carried out this assay using a 2:1 proportion of cancer cells as regards immune THP-1 cells. Assays were performed after 24 and 48 h of treatment and using 100 μM doses of selected compounds.

The results in Table 6 demonstrate that the effect on cancer cell viability was higher after 48 h of treatment and did not depend on the proportion of cancer and immune cells. We found that in the presence of an excess of THP-1 cells, MCF-7 viability decreased from 50% to 28% when cells were treated with *p*-bromo derivative **3**. The effect of cancer cell treatment with **5** and **7** hardly varied over time and remained between 40 and 50% of cancer cell viability. On the other hand, when MCF-7 cells were in a higher proportion than THP-1 cells, the effect of the derivatives was lower, but after 48 h of treatment, 50% of cancer cell inhibition was still found for compound **3,** and 30–40% inhibition was found for **5** and **7**, respectively.

#### 2.2.6. Study of the Effect of Selected Triazols on the Secretion of IL-6 to the Media

We determined the effect of these compounds on the secretion of IL-6 in monocultures of MCF-7 cells, to establish the effect on cancer cells in monocultures of immune cells THP-1 and in co-cultures of MCF-7 and THP-1 (1:5 proportion). Assays were performed by ELISA using cell media after treatments with selected compounds at 100 μM for 24 h.

The results in Table 7 show that the tested compounds decreased the secretion of IL-6 in MCF-7 cell line monocultures as the reference compound BMS-8 did. On the other hand, compounds enhanced the secretion of this target in monocultures of THP-1, and this effect was maintained when MCF-7 cells were co-cultured with THP-1 monocytes.

## 3. Discussion

Here, we describe the synthesis and biological evaluation of new entities bearing a triazole core as shown in Figure 3. In a versatile way, we synthetized fourteen derivatives bearing electron-withdrawing or -donating groups in R attached to a phenyl ring (see Figure 3) combined with a small and polar group (OH, P=H) or a bulky lipophilic one (P = 4,4′-dimetoxytrityl, see Figure 3). First, we determined the effect on cancer cell viability in monocultures, and then it was determined when co-cultured with immune cells Jurkat T or THP-1. In addition, we studied the activity of these compounds as PD-L1 and c-Myc downregulators, together with their effect on the distribution of PD-L1 and VEGFR-2 in cell membrane and on IL-6 secretion to the media.

As shown in the results description and in Table 4 and Table 6, all compounds were not active in inhibiting cell viability or proliferation in monocultures, but they were able to inhibit cancer cell viability in co-cultures with immune Jurkat T cells or THP-1 cell lines. This study was performed separately for three cancer cells lines: human HT-29, MCF-7, and A-549. One of the most outstanding results is that in general, neither of the compounds had a negative effect on Jurkat T nor THP-1 cell proliferation.

To establish correlations between the effect on the targets and the activation of the immune cells towards the tumor cells, in Figure 4, we have represented the effect of all bromine and methoxy derivatives **3–14** on each of the targets and the viability of tumor cells in the presence of T cells for the three cell lines HT-29 (a), MCF-7 (b), and A-549 (c). As regards the effect on HT-29, we can say that in general, there is a good correlation between the inhibition of cancer cell viability and total c-Myc and total PD-L1, with protected derivatives being more active than deprotected ones.

This correlation fits better for bromo than for methoxy compounds. That is, with derivatives **3–8**, bearing electron-withdrawing substituents, cell viability becomes lower; with compounds **5** or **7**, so do total PD-L1 and c-Myc, with inhibition rates from moderate to good. Better results were observed on the MCF-7 cell line. In this case, the correlation between the effect on cancer cell viability by the promotion of T cells and the effect on the targets is clearer. Again, bromo derivatives fit better in this correlation than methoxy ones, and we can highlight compounds **3**, **5,** and **7** because of their higher effect on this cell line. Contrary to that observed for HT-29 and MCF-7, in the case of A-549, the correlation is unclear, and this could be attributed to the very mild effect of the compounds in this particular cell line.

From this comparative analysis, we can assume that our compounds, and particularly bromo derivatives **3**, **5,** and **7**, are able to regulate not only PD-L1 but also c-Myc expression. This effect on the inhibition of both c-Myc and PD-L1 has been previously observed and described in the literature for other small molecules. For example, THZ1 [35] or JQ1 [36] reverse PD-L1-mediated immune escape by inhibiting the expression of c-Myc and open the door to more effective therapies to be used in combination with clinical anti-PD-L1 or anti-PD-1 mAbs.

In addition, when we study the effect of the most active derivatives, **3**, **5**, and **7,** on cancer cell viability in MCF-7 in the presence of THP-1 monocytes, we observed that these compounds were active even when the proportion of immune cells was very low (see Figure 5). In general, in these kinds of co-culture assays, we used a 1:5 proportion between cancer and immune cells, which is an excess of immune cells. This 1:5 proportion was the one used in Jurkat co-cultured assays, but in MCF-7 /THP-1 co-cultures, we tried both a 1:5 and 2:1 proportion. In this case, we still observed a good effect of the compounds on cancer cell viability in the presence of a lower proportion of THP-1 monocytes (2:1).

Finally, as regards the effect of **3**, **5** and **7** on IL-6 secretion to the media, we found that these compounds decreased the secretion of IL-6 in MCF-7 cell line monocultures as reference compound BMS-8 did. On the other hand, these compounds enhanced the secretion of IL-6 in monocultures of THP-1, and this effect was maintained when MCF-7 cells were co-cultured with THP-1 monocytes.

In conclusion, we have found compounds that are capable of influencing and altering the distribution of PD-L1 in tumor cells, and are thus capable of influencing the behavior of defensive cells towards cancer cells. In general, the MCF-7 and HT-29 cell lines are more sensitive to the oncoimmunomodulation action of the compounds than the A-549 cell line. In addition, triazole derivatives bearing the electron-withdrawing group in R and the bulky lipophilic group in P (see Figure 3) are the most active as PD-L1 and c-Myc downregulators and as potential immunomodulator agents. Moreover, compound **3** best combines the biological activity in all the targets and exhibits an interesting action on inflammation-related cytokine IL-6, which deserves to be further studied in the future.

## 4. Materials and Methods

### 4.1. Chemistry

#### 4.1.1. General Procedures

Spectra of 1H and 13C NMR were measured at 25 °C. The signals of the deuterated solvent (DMSO-d6) were taken as a reference. Multiplicity assignments of ^13^C signals were made by means of the DEPT pulse sequence. Complete signal assignments in ^1^H and ^13^C NMR spectra were made with the aid of 2D homo- and heteronuclear pulse sequences (COSY, HSQC, HMBC). High-resolution mass spectra were recorded using electrospray ionization–mass spectrometry (ESI–MS). Experiments that required an inert atmosphere were carried out under dry N_2_ in oven-dried glassware. Commercially available reagents were used as received.

#### 4.1.2. Experimental Procedure for the Synthesis of 2,2′-(1,4-Phenylene)bis(ethan-1-ol) **16**

A solution of 2,2′-(1,4-phenylene)diacetic acid **15** (1 eq) in anhydrous THF (20 mL) was cooled at 0 °C, and BH_3_·SMe_2_ (3 eq) was added dropwise under nitrogen atmosphere. The mixture was warmed up to room temperature and stirred for 12 h. Then, aqueous HCl 1 M solution was added to the solution, and the organic phase was extracted with AcOEt (3 × 50 mL). The collected organic phases were dried with anhydrous MgSO_4_. After filtration and evaporation of the solvent, compound **16** (97%) was obtained.

#### 4.1.3. Experimental Procedure for the Synthesis of 2-(4-(2-(Bis(4-methoxyphenyl)(phenyl)methoxy)ethyl)phenyl)ethan-1-ol **17**

Et_3_N (0,3 mL/mmol) and DMAP (0,05eq) were added to a solution of compound **16** (1,1 eq) in anhydrous CH_2_Cl_2_ (2,5 mL/mmol of **6**). Then, a solution of 4,4′-dimetoxytrityl chloride (1 eq) in anhydrous CH_2_Cl_2_ (1 mL/mmol) was added dropwise with a syringe pump. The resulting mixture was stirred at room temperature under a nitrogen atmosphere for 22 h. Then, the mixture was poured into brine and washed with brine (x3). The organic phase was dried with anhydrous MgSO_4_. After filtration and evaporation of the solvent, the residue was purified by column chromatography on silica gel as the stationary phase and a mixture of Hexanes:AcOEt (4:6, 1% Et_3_N) as the eluent to afford compound **17** (35%).

#### 4.1.4. Experimental Procedure for the Synthesis of 4-(2-(Bis(4-methoxyphenyl)(phenyl)methoxy)ethyl)phenethyl 4-Methylbenzenesulfonate **18**

DMAP and Et_3_N (1,3eq) were added to a solution of compound **17** (1 eq) in CH_2_Cl_2_ (5 mL/mmol). Then, TsCl (1,1 eq) was added dropwise, and the mixture was stirred at room temperature under a nitrogen atmosphere for 4 h. Then, the mixture was poured into a saturated aqueous solution of NH_4_Cl and extracted with CH_2_Cl_2_ (x3). The collected organic phases were dried with anhydrous MgSO_4_. After filtration and evaporation of the solvent, the residue was purified by column chromatography on silica gel as the stationary phase and a mixture of Hexanes:AcOEt (9:1, 8:2) as the eluent, affording compound **18** (62%).

#### 4.1.5. Experimental Procedure for the Synthesis of 4,4’-((4-(2-Azidoethyl)phenethoxy)(phenyl)methylene)bis(methoxybenzene) **19**

NaN_3_ (2 eq) was added to a solution of compound **18** (1 eq) in DMF (2,6 mL/mmol) The mixture was stirred at 50 °C under a nitrogen atmosphere for 3 h. Then, AcOEt was added, and the mixture was washed with brine (×3). The organic phase was dried with anhydrous MgSO_4_. After filtration and evaporation of the solvent, the residue was purified by column chromatography on silica gel as the stationary phase and a mixture of Hexanes:AcOEt (9:1, 8:2) as the eluent, affording compound **19** (83%).

#### 4.1.6. Experimental Procedure for the Synthesis of the Non-Commercially Available Alkynes **21**–**26**

Kristensen modification of the Ohira–Bestmann reaction was used for the synthesis of alkynes [37]. Thus, dimethyl-2-oxopropylphosphonate (1,2 eq) was added dropwise to a suspension of K_2_CO_3_ (4,5 eq) and 4-acetamidobenzenesulfonyl azide (ABSA) (1,3 eq) in MeCN (10 mL/mmol). The colorless suspension was stirred at room temperature for 2 h. Then, a solution of the corresponding aldehyde (1 eq) in MeOH (10 mL/mmol) was added to the now pale-yellow suspension and stirred at room temperature under a nitrogen atmosphere for 15 h. Finally, the reaction mixture was filtered over celite, and the filter was thoroughly washed with diethyl ether (x3). After filtration and evaporation of the solvent, the residue was purified by column chromatography on silica gel as the stationary phase and Hexanes as the eluent, affording alkynes **21–26**.

#### 4.1.7. Experimental Procedure for the Synthesis of Triazole Derivates **1**, **3**, **5**, **7**, **9**, **11** and **13**

Compound **19** (1 eq) and the corresponding alkyne (1,2 eq) were dissolved in DMF, and on the other hand, CuSO_4_·5H_2_O (0,1 eq) and ascorbic acid sodium salt (0,1 eq) were dissolved in water, so that the DMF/H_2_O ratio was 9:1, respectively (20 mL/mmol). The water solution was added dropwise over the DMF solution, and the resulting mixture was stirred at 60 °C under a nitrogen atmosphere for 2 h. Then, AcOEt was added to the mixture and washed with brine (×3). The organic phase was dried with anhydrous MgSO_4_. After filtration and evaporation of the solvent, the residue was purified by column chromatography on silica gel as the stationary phase and a mixture of Hexanes:AcOEt (8:2, 7:3, 1.1, 3:4) as the eluent, affording compounds **1**, **3**, **5**, **7**, **9**, **11** and **13**.

#### 4.1.8. Experimental Procedure for the Deprotection of Triazole Derivates

CF_3_COOH (0,08 mL/mmol) was added dropwise to a solution of the corresponding triazole derivative (1 eq) in MeOH (18,2 mL/mmol). The mixture was stirred at room temperature under a nitrogen atmosphere for 30 min. Then, a NaHCO_3_ aqueous solution (2,7 eq) was added to the mixture. After filtration and evaporation of the solvent, the residue was purified by column chromatography on silica gel as the stationary phase and a mixture of Hexanes:AcOEt (6:4, 1.1, 3:4) as the eluent, affording compounds **2**, **4**, **6**, **8**, **10**, **11** and **14**.

**1-bromo-4-ethynylbenzene 21**: yield 42%; white solid; m.p. 74.5–75.0 °C; ^1^H NMR (400 MHz, DMSO-d6) δ 7.59 (d, J = 8.5 Hz, 2H), 7.43 (d, J = 8.5 Hz, 2H), 4.29 (s, 1H); ^13^C NMR (100 MHz, DMSO-d6); δ 133.60 (2xCH), 131.71 (2xCH), 122.38 (C), 120.92 (C), 82.34 (C), 82.06 (CH).

**1-bromo-3-ethynylbenzene 22:** yield 53%; colorless liquid; ^1^H NMR (400 MHz, DMSO-d6) δ 7.68 (s, J = 6.1 Hz, 1H), 7.63 (d, J = 7.7 Hz, 1H), 7.49 (dt, J = 7.9 Hz, J = 1.2 Hz, 1H), 7.35 (t, J = 7.9 Hz, 1H), 4.32 (s, 1H); ^13^C NMR (100 MHz, DMSO-d6); δ 133.90 (CH), 132.01 (CH), 130.76 (CH), 130.72 (CH), 123.90 (C), 121.56 (C), 82.29 (C), 81.77 (CH).

**1-bromo-2-ethynylbenzene 23:** yield 67%; colorless liquid; ^1^H NMR (400 MHz, DMSO-d6) δ 7.71 (dd, J = 7.9 Hz, J = 1.2 Hz, 1H), 7.60 (dd, J = 7.6 Hz, J = 1.8 Hz, 1H), 7.41 (dt, J = 7.5 Hz, J = 1.3 Hz, 1H), 7.35 (dt, J = 7.7 Hz, J = 1.8 Hz, 1H), 4.55 (s, 1H); ^13^C NMR (100 MHz, DMSO-d6); δ 133.98 (CH), 132.42 (CH), 130.65 (CH), 127.75 (CH), 124.69 (C), 123.59 (C), 85.36 (C), 81.71 (CH).

**1-ethynyl-4-methoxybenzene 24:** yield 40%; white solid; colorless liquid; ^1^H NMR (400 MHz, DMSO-d6) δ 7.41 (d, J = 8.9 Hz, 2H), 6.93 (d, J = 8.9 Hz, 2H), 3.99 (s,1H), 3.77 (s,3H); 13C NMR (100 MHz, DMSO-d6) δ 159.57 (C), 133.17 (2xCH), 114.25 (2xCH), 113.64 (C), 83.48 (C), 79.08 (CH), 55.21 (CH_3_).

**1-ethynyl-3-methoxybenzene 25:** yield 57%; colorless liquid; ^1^H NMR (400 MHz, DMSO-d6) δ 7.29 (t, J = 7.9 Hz, 1H), 7.05 (dt, J = 7.5 Hz, J = 1.1 Hz, 1H), 7.00–6.95 (m, 2H), 4.15 (s,1H), 3.76 (s,3H); ^13^C NMR (100 MHz, DMSO-d6) δ 159.09 (C), 129.81 (CH), 124.00 (CH), 122.76 (C), 116.53 (CH), 115.35 (CH), 83.35 (C), 79.52 (CH), 55.18 (CH_3_).

**1-ethynyl-2-methoxybenzene 26:** yield 52%; white solid; colorless liquid; ^1^H NMR (400 MHz, DMSO-d6) δ 7.40 (dt, J = 7.5 Hz, J = 2.1 Hz, 1H), 7.36 (dd, J = 7.1 Hz, J = 1.3 Hz,1H), 7.05 (d, J = 8.1 Hz, 1H), 6.93 (dt, J = 7.5 Hz, J = 1.0 Hz, 1H), 4.20 (s,1H), 3.81 (s,3H);^13^C NMR (100 MHz, DMSO-d6) δ 160.21 (C), 133.44 (CH), 130.42 (CH), 120.34 (CH), 111.29 (CH), 110.69 (C), 84.36 (C), 80.13 (CH), 55.51 (CH_3_).

**2,2′-(1,4-phenylene)bis(ethan-1-ol) 16:** yield 97%; white solid; m.p. 75–82 °C; ^1^H NMR (400 MHz, MeOH-d4); δ 7.14 (s, 4H), 4.77 (s, 2H), 3.72 (t, J = 7.1 Hz, 4H), 2.78 (t, J = 7.1 Hz, 4H); ^13^C NMR (100 MHz, MeOH-d4) δ 138.1 (2xC), 129.9 (2xCH), 64.3 (2xCH_2_), 39.8 (2xCH_2_).

**2-(4-(2-(bis(4-methoxyphenyl)(phenyl)methoxy)ethyl)phenyl)ethan-1-ol 17:** yield 35%; yellow oil; ^1^H NMR (400 MHz, DMSO-d6) δ 7.30–7.07 (m, 13H), 6.85 (d, J = 8.9 Hz, 4H), 4.58 (t, J = 5.2 Hz, 1H), 3.72 (s, 6H), 3.57 (c, J = 7.1 Hz, 2H), 3.10 (t, J = 6.8 Hz, 2H), 2.78 (t, J = 6.8 Hz, 2H), 2.67 (t, J = 7.1 Hz, 2H); ^13^C NMR (100 MHz, DMSO-d6) δ 157.93 (2xC), 145.02 (C), 137.13 (C), 136.55 (C), 135.86 (2xC), 129.53 (4xCH), 128.74 (2xCH), 128.64 (2xCH), 127.70 (2xCH), 127.60 (2xCH), 126.51 (CH), 113.07 (4xCH), 85.37 (C), 64.65 (CH_2_), 62.25 (2xCH_2_), 54.96 (2xCH_3_), 35.51 (CH_2_).

**4-(2-(bis(4-methoxyphenyl)(phenyl)methoxy)ethyl)phenethyl-4-methylbenzenesulfonate 18:** yield 62%; colorless oil; ^1^H NMR (400 MHz, DMSO-d6) δ 7.65 (d, J = 8.3 Hz, 2H), 7.35 (d, J = 7.9 Hz, 2H), 7.30–7.15 (m, 9H), 7.06 (d, J = 8.9 Hz, 4H), 6.84 (d, J = 9.0 Hz, 4H), 4.20 (t, J = 6.5 Hz, 2H), 3.72 (s, 6H), 3.12 (t, J = 6.7 Hz, 2H), 2.85 (t, J = 6.5 Hz, 2H), 2.78 (t, J = 6.6 Hz, 2H), 2.36 (s, 3H); ^13^C NMR (100 MHz, DMSO-d6) δ 158.49 (2xC), 145.51 (C), 145.28 (C), 137.94 (C), 136.36 (2xC), 134.92 (C), 132.73 (C), 130.50 (2xCH), 130.04 (4xCH), 129.46 (2xCH), 129.06 (2xCH), 128.27 (2xCH), 128.13 (2xCH), 127.92 (2xCH), 127.08 (CH), 113.57 (4xCH), 85.93 (C), 71.59 (CH_2_), 64.92 (CH2), 54.47 (2xCH_3_), 36.01 (CH_2_), 34.43 (CH_2_), 21.48 (CH_3_).

**4,4’-((4-(2-azidoethyl)phenethoxy)(phenyl)methylene)bis(methoxybenzene) 19**: yield 83%; colorless oil; ^1^H NMR (400 MHz, DMSO-d6) δ 7.32–7.20 (m,13H), 6.83 (d, J = 8.9 Hz, 4H), 3.72 (s, 6H), 3.54 (t, J = 7.0 Hz, 2H), 3.12 (t, J = 6.7 Hz, 2H), 2.80 (m, 4H); ^13^C NMR (100 MHz, DMSO-d6) δ 158.0 (2xC), 145.1 (C), 137.1 (C), 136.0 (C), 135.8 (2xC), 129.5 (4xCH), 129.0 (2xCH), 128.4 (2xCH), 127.6 (4xCH), 126.4 (CH), 113.1 (4xCH), 85.2 (C), 64.6 (CH_2_), 55.0 (CH_2_), 51.5 (2xCH_3_), 35.5 (CH2), 33.9 (CH_2_).

**1-(4-(2-(bis(4-methoxyphenyl)(phenyl)methoxy)ethyl)phenethyl)-4-phenyl-1*H*-1,2,3-triazole 1:** yield 66%; colorless oil; ^1^H NMR (400 MHz, DMSO-d6) δ 8.49 (s, 1H), 7.77 (d, J = 7.0 HZ, 2H), 7.41 (t, 2H), 7.20 (m, 14H), 6.83 (d, J = 9.0 Hz, 4H), 4.63 (t, 2H), 3.72 (s, 6H), 3.19 (t, 2H), 3.11 (t, 2H), 2.77 (t, 2H);^13^C NMR (100 MHz, DMSO_d6_) δ 157.98 (C), 146.14 (C), 145.02 (C), 137.42 (C), 135.89 (2xC), 135.32 (C), 130.80 (2xC) 129.57 (4xCH), 129.08 (2xCH), 128.85 (2xCH), 128.50 (2xCH), 127.74 (2xCH), 127.62 (CH), 126.54 (CH), 125.06 (4xCH), 121.22 (CH), 113.10 (4xCH), 85.41 (C), 64.39 (CH_2_), 55.00 (2xCH_3_), 50.65 (CH_2_), 35.51 (CH_2_), 35.19 (CH_2_).

**2-(4-(2-(4-phenyl-1*H*-1,2,3-triazol-1-yl)ethyl)phenyl)ethan-1-ol 2:** yield 46%; colorless oil; ^1^H NMR (400 MHz, DMSO-d6) δ 8.51 (s,1H), 7.80 (d, J = 7.1 Hz, 2H), 7.44 (t, 2H), 7.32 (t, 1H), 7.13 (s, 4H), 4.63 (t, 2H), 3.56 (t, 2H), 3.33 (s, 1H), 3.18 (t,2H), 2.67 (t, 2H); ^13^C NMR (100 MHz, DMSO-d6) δ 146.11 (C), 137.76 (C), 134.99 (C), 130.81 (C), 128.92 (2xCH), 128.86 (2xCH), 128.44 (2xCH), 127.75 (CH), 125.05 (2xCH), 121.24 (CH), 62.09 (CH_2_), 50.65 (CH_2_), 38.60 (CH_2_), 35.16 (CH_2_); HR ESMS *m*/*z* 294.1606 [M-H]+. Calc. for C_18_H_19_N_3_O 294.1606

**1-(4-(2-(bis(4-methoxyphenyl)(phenyl)methoxy)ethyl)phenethyl)-4-(4-bromophenyl)-1*H*-1,2,3-triazole 3:** yield 56%; colorless oil; ^1^H NMR (400 MHz, DMSO-d6) δ 8.54 (s,1H), 7.72 (d, J = 8.6 Hz, 2H), 7.59 (d, J = 8.7 Hz, 2H), 7.26–7.11 (m, 13H), 6.83 (d, J = 9.0 Hz, 4H), 4.64 (t, J = 7.3 Hz, 2H), 3.72 (s, 6H), 3.19 (t, J = 7.3 Hz, 2H), 3.10 (t, J = 6.7 Hz, 2H), 2.77 (t, J = 6.7 Hz, 2H); ^13^C NMR (100 MHz, DMSO-d6) δ 157.95 (2xC), 145.05 (C), 137.43 (C), 135.86 (2xC), 135.24 (C), 131.80 (2xCH), 130.02 (C), 129.54 (4xCH), 129.09 (CH), 128.48 (CH), 127.72 (CH), 127.59 (CH), 127.00 (2xCH), 126.52 (CH), 121.60 (CH), 120.67 (C), 113.07 (8xCH), 85.37 (C), 64.35 (CH_2_), 59.72 (C), 54.98 (2xCH_3_), 50.68 (CH_2_), 35.49 (CH_2_), 35.13 (CH_2_); HR ESMS *m*/*z* 696.1835 [M-Na]+. Calc. for C_39_H_36_N_3_O_3_Br 696.1833.

**2-(4-(2-(4-(4-bromophenyl)-1*H*-1,2,3-triazol-1-yl)ethyl)phenyl)ethan-1-ol 4:** yield 52%; white solid; m.p. 135–140 °C; ^1^H NMR (400 MHz, DMSO-d6) δ 8.57 (s,1H), 7.76 (d, J = 8.5 Hz, 2H), 7.64 (d, J = 8,5 Hz, 2H), 7.12 (s, 4H), 4.63 (t, 2H), 4.57 (t, 2H) 3.56 (m, 2H), 3.17 (t, 2H), 2.67 (t, 2H); ^13^C NMR (100 MHz, DMSO-d6) δ 145.04 (C), 137.77 (C), 134,91 (C), 131.82 (2xCH), 130.05 (C), 128.91 (2xCH), 128.40 (2xCH), 127.01 (2xCH), 121.62 (CH), 120.67 (C), 62,07 (CH_2_), 50,68 (CH_2_), 35.10 (CH_2_), 30.68 (CH_2_); HR ESMS *m*/*z* 372.0710 [M-H]+. Calc. for C_18_H_18_N_3_OBr 372.0711.

**1-(4-(2-(bis(4-methoxyphenyl)(phenyl)methoxy)ethyl)phenethyl)-4-(3-bromophenyl)-1*H*-1,2,3-triazole 5:** yield 70%; colorless oil; ^1^H NMR (400 MHz, DMSO-d6) δ 8.53 (s,1H), 7.99 (t, J = 1.7 Hz, 1H), 7.78 (dt, J = 7.9 Hz, J = 1.3 Hz, 1H), 7.51 (dd, J = 8.0 Hz, J = 2.0 Hz, 1H), 7.37 (t, J = 7.9 Hz, 1H), 7.26–7.09 (m, 13H), 6.84 (d, J = 8.9 Hz, 4H), 4.64 (t, J = 7.3 Hz, 2H), 3.72 (s, 6H), 3.19 (t, J = 7.3 Hz, 2H), 3.10 (t, J = 6.7 Hz, 2H), 2.77 (t, J = 6.7 Hz, 2H);^13^C NMR (100 MHz, DMSO-d6) δ 157.96 (2xC), 145.03 (C), 144.67 (C),137.43 (C), 135.86 (2xC), 135.22 (C), 133.13 (C), 131.10 (CH), 130.41 (CH), 129.56 (4xCH), 129.09 (CH), 128.49 (CH), 127.73 (CH), 127.60 (CH), 127.49 (CH), 126.52 (CH), 123.93 (CH), 122.24 (C), 121.99 (CH), 113.08 (8xCH), 85.38 (C), 64.37 (CH_2_), 54.99 (2xCH_3_), 50.74 (CH_2_), 35.50 (CH_2_), 35.07 (CH_2_); HR ESMS *m*/*z* 696.1835 [M-Na]+. Calc. for C_39_H_36_N_3_O_3_Br 696.1833.

**2-(4-(2-(4-(3-bromophenyl)-1*H*-1,2,3-triazol-1-yl)ethyl)phenyl)ethan-1-ol 6:** yield 64%; white solid; m.p. 129–131 °C; ^1^H NMR (400 MHz, DMSO-d6) δ 8.63 (s,1H), 7.99 (s,1H), 7.82 (d, J = 7.8 Hz, 1H), 7.51 (d, J = 8.8 Hz, 1H), 7.41 (t, 1H), 7.12 (s, 4H), 4.62 (t, 2H), 4.57 (t, 1H) 3.56 (m, 2H), 3.17 (t, 2H), 2.66 (t, 2H); ^13^C NMR (100 MHz, DMS = -d6) δ 144.63 (C), 137.77 (C), 134,89 (C), 133.13 (CH), 131.12 (CH), 130.41 (C), 128.91 (2xCH), 128.40 (2xCH), 127.01 (CH), 123.91 (CH), 122.21 (CH), 121.98 (C), 62,07 (CH_2_), 50,74 (CH_2_), 38.59 (CH_2_), 35.05 (CH2); HR ESMS *m*/*z* 372.0708 [M-H]+. Calc. for C_18_H_18_N_3_OBr 372.0711.

**1-(4-(2-(bis(4-methoxyphenyl)(phenyl)methoxy)ethyl)phenethyl)-4-(2-bromophenyl)-1*H*-1,2,3-triazole 7:** yield 40%; colorless oil; ^1^H NMR (400 MHz, DMSO-d6) δ 8.49 (s,1H), 7.87 (dd, J = 7.8 Hz, J = 1.7 Hz, 1H), 7.70 (dd, J = 8.1 Hz, J = 1.1 Hz, 1H), 7.46 (dt, J = 7.6 Hz, J = 1.3 Hz, 1H), 7.30–7.20 (m, 14H), 6.83 (d, J = 9.0 Hz, 4H), 4.68 (t, J = 7.3 Hz, 2H), 3.72 (s, 6H), 3.18 (t, J = 7.2 Hz, 2H), 3.11 (t, J = 6.8 Hz, 2H), 2.77 (t, J = 6.8 Hz, 2H); ^13^C NMR (100 MHz, DMSO-d6) δ 157.95 (2xC), 145.03 (C), 143.85 (C),137.33 (C), 135.87 (2xC), 135.27 (C), 133.42 (CH), 131.38 (C), 130.32 (CH), 129.67 (CH), 129.56 (4xCH), 129.05 (CH), 128.58 (CH), 127.92 (CH), 127.73 (CH), 127.61 (CH), 126.53 (CH), 124.01 (CH), 120.58 (C), 113.09 (8xCH), 85.41 (C), 64.44 (CH_2_), 54.98 (2xCH_3_), 50.64 (CH_2_), 35.51 (CH_2_), 35.38 (CH_2_); HR ESMS *m*/*z* 696.1835 [M-Na]+. Calc. for C_39_H_36_N_3_O_3_Br 696.1833.

**2-(4-(2-(4-(2-bromophenyl)-1*H*-1,2,3-triazol-1-yl)ethyl)phenyl)ethan-1-ol 8:** yield 75%; white solid; m.p. 130–133 °C; ^1^H NMR (400 MHz, DMSO-d6) δ 8.51 (s,1H), 7.89 (d, J = 7.8 Hz, 1H), 7.72 (d, J = 8.0 Hz, 1H), 7.48 (t, 1H), 7.30 (t, 1H), 7.12 (s, 4H), 4.68 (t, 2H), 4.50 (t, 1H), 3.56 (m, 2H), 3.17 (t, 2H), 2.67 (t, 2H); ^13^C NMR (100 MHz, DMSO-d6) δ 143.87 (C), 137.74 (C), 134,99 (C), 133.46 (CH), 131.39 (C), 130.35 (C), 129.71 (CH), 128.94 (2xCH), 128.54 (2xCH), 127.97 (CH), 124.05 (CH), 120.59 (CH), 62.17 (CH_2_), 50.69 (CH_2_), 38.64 (CH_2_), 35.41 (CH_2_); HR ESMS *m*/*z* 372.0716 [M-H]+. Calc. for C_18_H_18_N_3_OBr 372.0711.

**1-(4-(2-(bis(4-methoxyphenyl)(phenyl)methoxy)ethyl)phenethyl)-4-(4-methoxyphenyl)-1*H*-1,2,3-triazole 9:** yield 54%; yellow oil; ^1^H NMR (400 MHz, DMSO-d6) δ 8.38 (s,1H), 7.69 (d, J = 8.8 Hz, 2H), 7.29–7.09 (m,13H), 6.97 (d, J = 8.9 Hz, 2H), 6.83 (d, J = 8.9 Hz,4H), 4.61 (t, J = 7.3 Hz, 2H), 3.77 (s, 3H), 3.72 (s, 6H), 3.18 (t, J = 7.3 Hz, 2H), 3.10 (t, J = 6.7 Hz, 2H), 2.77 (t, J = 6.6 Hz, 2H); ^13^C NMR (100 MHz, DMSO-d6) δ 158.92 (C), 157.96 (2xC), 146.07 (C), 145.03 (C), 137.41 (C), 135.88 (2xC), 135.36 (C), 129.56 (6xCH), 129.09 (CH), 128.51 (CH), 127.74 (CH), 127.61 (CH), 126.40 (2xCH), 123.41 (CH), 120.28 (CH), 114.27 (2xCH), 113.10 (6xCH), 85.39 (C), 64.40 (CH_2_), 59.75 (C), 55.13 (CH_3_), 54.99 (2xCH_3_), 50.58 (CH_2_), 35.51 (CH_2_), 35.21 (CH_2_); HR ESMS *m*/*z* 648.2838 [M-Na]+. Calc. for C_40_H_39_N_3_O_4_ 648.2835.

**2-(4-(2-(4-(4-methoxyphenyl)-1*H*-1,2,3-triazol-1-yl)ethyl)phenyl)ethan-1-ol 10:** yield 20%; white solid; m.p. 172–174 °C; ^1^H NMR (400 MHz, DMSO-d6) δ 8.40 (s, 1H), 7.72 (d, J = 8.9 Hz, 2H), 7.12 (s, 4H), 7.00 (d, J = 8.9 Hz, 2H), 4.62–4.57 (m, 3H), 3.78 (s, 3H), 3.56 (m, 2H), 3.17 (t, 2H), 2.67 (t, 2H); ^13^C NMR (100 MHz, DMSO-d6) δ 158.93 (C), 146.04 (C), 137.75 (C), 135.03 (C), 128.92 (2xCH), 128.43 (2xCH), 126.40 (2xCH), 123.43 (C), 120.29 (CH), 114.29 (2xCH), 62.10 (CH_2_), 55.13 (CH_3_), 50.58 (CH_2_), 38.60 (CH_2_), 35.18 (CH_2_); HR ESMS *m*/*z* 324.1710 [M-H]+. Calc. for C_19_H_21_N_3_O_2_ 324.1712.

**1-(4-(2-(bis(4-methoxyphenyl)(phenyl)methoxy)ethyl)phenethyl)-4-(3-methoxyphenyl)-1*H*-1,2,3-triazole 11:** yield 74%; colorless oil; ^1^H NMR (400 MHz, DMSO-d6) δ 8.53 (s,1H), 7.38–7.07 (m, 16H), 6.9–6.82 (m, 5H), 4.63 (t, J = 7.3 Hz, 2H), 3.79 (s, 3H), 3.71 (s, 6H), 3.19 (t, J = 7.3 Hz, 2H), 3.11 (t, J = 6.7 Hz, 2H), 2.77 (t, J = 6.6 Hz, 2H); ^13^C NMR (100 MHz, DMSO-d6) δ 159.69 (C), 157.99 (2xC), 146.08 (C), 145.05 (C), 137.44 (C), 135.91 (2xC), 135.32 (C), 132.16 (C), 130.03 (CH), 129.59 (4xCH), 129.11 (2xCH), 128.53 (2xCH), 127.76 (2xCH), 127.64 (2xCH), 126.56 (CH), 121.49 (CH), 117.45 (CH), 113.53 (CH), 113.11 (4xCH), 110.30 (CH), 85.43 (C), 64.43 (CH_2_), 55.10 (CH_3_), 55.00 (2xCH_3_), 50.69 (CH_2_), 35.54 (CH_2_), 35.20 (CH_2_); HR ESMS *m*/*z* 648.2838 [M-Na]+. Calc. for C_40_H_39_N_3_O_4_ 648.2835.

**1-(4-(2-(bis(4-methoxyphenyl)(phenyl)methoxy) 2-(4-(2-(4-(3-methoxyphenyl)-1*H*-1,2,3-triazol-1-yl)ethyl)phenyl)ethan-1-ol 12:** yield 55%; white solid; m.p. 90–93 °C; ^1^H NMR (400 MHz, DMSO-d6) δ 8.54 (s, 1H), 7.39–7.32 (m, 3H), 7.13 (s, 4H), 6.89 (d, J = 8.2 Hz, 1H), 4.62 (t, 2H), 4.57 (t, 1H), 3.81 (s, 3H), 3.56 (c, 2H), 3.18 (t, 2H), 2.67 (t, 2H); ^13^C NMR (100 MHz, DMSO-d6) δ 159.65 (C), 145.99 (C), 137.75 (C), 134.96 (C), 132.13 (C), 129.98 (CH), 128.91 (2xCH), 128.41 (2xCH), 121.47 (CH), 117.39 (CH), 113.50 (CH), 110.25 (CH), 62.07 (CH_2_), 55.07 (CH_3_), 50.64 (CH_2_), 38.59 (CH_2_), 35.13 (CH_2_); HR ESMS *m*/*z* 324.1716 [M-H]+. Calc. for C_19_H_21_N_3_O_2_ 324.1712.

**1-(4-(2-(bis(4-methoxyphenyl)(phenyl)methoxy)ethyl)phenethyl)-4-(2-methoxyphenyl)-1*H*-1,2,3-triazole 13:** yield 51%; colorless oil; ^1^H NMR (400 MHz, DMSO-d6) δ 8.27 (s,1H), 8.12 (dd, J=7.7 Hz, J = 1.7 Hz, 1H), 7.32–7.01 (m,16H), 6.84 (d, J = 9 Hz, 4H), 4.64 (t, J = 7.3 Hz, 2H), 3.82 (s, 3H), 3.71 (s, 6H), 3.17 (t, J=7.3 Hz, 2H), 3.11 (t, J = 6.8 Hz, 2H), 2.78 (t, J = 6.7 Hz, 2H); ^13^C NMR (100 MHz, DMSO_d6_) δ 157.99 (2xC), 155.30 (C), 145.05 (C), 141.58 (C), 137.38 (C), 135.90 (2xC), 135.48 (C), 129.59 (4xCH), 129.08 (2xCH), 128.81 (CH), 128.60 (2xCH), 127.77 (2xCH), 127.63 (2xCH), 126.54 (2xCH), 123.85 (CH), 120.62 (CH), 119.22 (C), 113.12 (4xCH), 111.54 (CH), 85.44 (C), 64.46 (CH_2_), 55.37 (CH_3_), 55.01 (2xCH_3_), 50.46 (CH_2_), 35.54 (CH_2_), 30.68 (CH_2_); HR ESMS *m*/*z* 648.2838 [M-Na]+. Calc. for C_40_H_39_N_3_O_4_ 648.2835.

**2-(4-(2-(4-(2-methoxyphenyl)-1*H*-1,2,3-triazol-1-yl)ethyl)phenyl)ethan-1-ol 14:** yield 37%; white solid; m.p. 76–80 °C; ^1^H NMR (400 MHz, DMSO-d6) δ 8.28 (s, 1H), 8.12 (dd, J = 7.7 Hz, 1.7Hz, 1H), 7.32 (t, 1H), 7.14 (s, 4H), 7.11 (dd, J = 8.4 Hz, 0.8 Hz, 1H), 7.04 (t, 1H), 4.64 (t, 2H), 4.59 (t, 1H), 3.88 (s, 3H), 3.57 (m, 2H), 3.17 (t, 2H), 2.68 (t, 2H); ^13^C NMR (100 MHz, DMS = -d6) δ 155.29 (C), 141.54 (C), 137.73 (C), 135.14 (C), 128.91 (2xCH), 128.78 (2xCH), 128.5 (CH), 126.51 (CH), 123.81 (CH), 120.60 (CH), 119.21 (C), 115.55 (CH), 62.12 (CH_2_), 55.41 (CH_3_), 50.43 (CH_2_), 38.62 (CH_2_), 35.38 (CH_2_); HR ESMS *m*/*z* 324.1711 [M-H]+. Calc. for C_19_H_21_N_3_O_2_ 324.1712.

### 4.2. Biological Studies

#### 4.2.1. Cell Culture

Cell culture media were purchased from Gibco (Grand Island, NY, USA). Fetal bovine serum (FBS) was obtained from Harlan-Seralab (Belton, UK). Supplements and other chemicals not listed in this section were obtained from Sigma Chemical Co. (St. Louis, MO, USA). Plastics for cell cultures were supplied by Thermo Scientific BioLite. All tested compounds were dissolved in DMSO at a concentration of 20 mM and stored at −20 °C until use.

HT-29, A549, MCF-7, and HEK-293 cell lines were maintained in Dulbecco’s modified Eagle’s medium (DMEM) containing glucose (1 g/L), glutamine (2 mM), penicillin (50 μg/mL), streptomycin (50 μg/mL), and amphotericin B (1.25 μg/mL), supplemented with 10% FBS.

#### 4.2.2. Cell Proliferation Assay

In 96-well plates, 5 × 10^3^ (HT-29, MCF-7, A549, and HEK-293) cells per well were incubated with serial dilutions of the tested compounds in a total volume of 100 μL of their growth media. The 3-(4,5-dimethylthiazol-2-yl)-2,5-diphenyltetrazolium bromide (MTT; Sigma Chemical Co.) dye reduction assay in 96-well microplates was used. After 2 days of incubation (37 °C, 5% CO_2_ in a humid atmosphere), 10 μL of MTT (5 mg/mL in phosphate-buffered saline, PBS) was added to each well, and the plate was incubated for a further 3 h (37 °C). After that, the supernatant was discarded and replaced by 100 µL of DMSO to dissolve formazan crystals. The absorbance was then read at 550 nm by spectrophotometry. For all concentrations of the compound, cell viability was expressed as the percentage of the ratio between the mean absorbance of treated cells and the mean absorbance of untreated cells. Three independent experiments were performed, and the IC_50_ values (i.e., concentration inhibiting half of cell proliferation) were graphically determined using GraphPad Prism 4 software.

#### 4.2.3. PD-L1 and c-Myc Relative Quantification by Flow Cytometry

To study the effect of the compounds on every biological target in cancer cell lines, the compounds were used at a 100 µM dose.

For the assay, 10^5^ cells per well were incubated for 24 h with the corresponding dose of the tested compound in a total volume of 500 μL of their growth media.

To detect total PD-L1 and c-Myc, after the cell treatments, they were collected and fixed with 4% in PBS paraformaldehyde. After fixation, a treatment with 0,5% in PBS Triton^™^ X-100 was performed, and finally, cells were stained with FITC Mouse monoclonal Anti-c-Myc (ab223913) and Alexa Fluor^®^ 647 Rabbit monoclonal Anti-PD-L1 (ab215251).

#### 4.2.4. Cell Viability Evaluation in Co-Cultures

To study the effect of the compounds on the cell viability in co-culture with Jurkat T cells or THP-1 cells, 10^5^ or 2 × 10^5^ of the corresponding cancer cells line per well were seeded and incubated for 24 h, then the medium was changed by one cell culture medium supplemented with IFN-γ (10 ng/mL; human, Invitrogen^®^) and containing 5 × 10^5^ or 10^5^ Jurkat T cells or THP-1, respectively, per well and 100 µM of the corresponding compound or DMSO for the positive control. After 24 h/48 h of incubation, supernatants were collected to determine Jurkat T or THP-1 living cells, and on the other hand, corresponding cancer cells were collected with trypsin. Both types of suspension cells were fixed with 4% in PBS paraformaldehyde and counted by flow cytometry.

#### 4.2.5. PD-L1 and VEGFR-2 Relative Quantification by Flow Cytometry in Co-Cultures

To study the effect of the compounds on every biological target in co-cultured cancer cell lines, the compounds were incubated for 24h as described before.

To detect membrane PD-L1 and VEGFR-2, after the cell treatments, they were collected, fixed with 4% in PBS paraformaldehyde, and stained with FITC Mouse monoclonal Anti-Human VEGFR-2 (ab184903) and Alexa Fluor^®^ 647 Rabbit monoclonal Anti-PD-L1 (ab215251).

#### 4.2.6. Fluorescence Microscopy

For immunofluorescence microscopy of the microtubule network, 106 cells were plated on cover glass and incubated with the selected compounds. Cells were then fixed in 4% formaldehyde (in PBS pH 7.4) for 20 min at room temperature and permeabilized with PBS-Triton X-100 0.5% for 1 min at room temperature. Direct immunostaining was carried out for 1 h at room temperature with FITC Mouse monoclonal Anti-c-Myc (ab223913) and Alexa Fluor^®^ 647 Rabbit monoclonal Anti-PD-L1 (ab215251). Then, cells were washed in PBS, and cover glasses were mounted with a drop of ProLong^®^ antifade solution (Invitrogen). The cytoskeleton was imaged by a confocal laser scanning microscope (CLSM) Leica SP5 with a Leica inverted microscope, equipped with 20x objective. Each image was recorded with the CLSM’s spectral mode selecting specific domains of the emission spectrum. The FITC fluorophore was excited at 488 nm with an argon laser, and its fluorescence emission was collected between 496 nm and 535 nm.

#### 4.2.7. RT-qPCR Assay

MCF-7 cells at 70–80% confluence were collected, and 1.5 × 105 cells were placed in a six-well plate in 1.5 mL of medium. After 24h, cells were incubated with the corresponding compounds for 48 h. Cells were collected, and the total cellular RNA from MCF-7 cells was isolated using the Ambion RNA extraction kit according to the manufacturer’s instructions. The cDNA was synthesized by MMLV-RT with 1–21 µg of extracted RNA and oligo(dT)15 according to the manufacturer’s instructions. Genes were amplified by a thermal cycler and StepOnePlus^™^ TaqMan^®^ probes. TaqMan^®^ Gene Expression Master Mix Fast containing the appropriate buffer for the amplification conditions, dNTPs, thermostable DNA polymerase enzymes, and a passive reference probe was used. To amplify each of the genes, the predesigned primers sold by Life Technologies TaqMan^®^ Gene Expression Assays, Hs99999903-m1 (β-actin), Hs00153408-m1 (c-Myc), and Hs00204257-m1, (PD-L1) were used.

## Data Availability

Appendix A is available.

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
