# Peer review of "Synthesis and Biological Evaluation of Potential Oncoimmunomodulator Agents"

_ijms, 2023, doi:10.3390/ijms24032614_

Round 1

Reviewer 1 Report

Gil-Edo et al. report synthesis and biological evaluation of potential oncoimmunomodulator agents. The authors synthesized fourteen triazole-scaffold derivatives and evaluated them as potential oncoimmunomodultator agents by targeting both PD-L1 and c-Myc. The anti-cancer activity of these molecules was tested on monocultures of several tumor cell lines (HT-29, A-549, and 10 MCF-7) and on the non-tumor cell line HEK-293., The effect on cancer cell viability, when co-cultured with immune cells (Jurkat T cells or THP-1), has also been determined. 

I have some major concerns about this ms that need to be addressed before the publication.

1.      Invitro anti-cancer activity of these compounds is too low (IC50>300 mM). Hence, Authors should synthesize more derivatives to get better anti-cancer activity.

2.      Authors have used a very high concentration (100 mM) of the synthesized compound to assess efficacy against cancer cell death and n c-Myc and PD-L1 expression. This concentration is too high, so authors should include various concentrations of compounds (below 100mM) in experiments.

3.      Authors should also calculate IC50 against VEGFR2, PD-L1, and c-Myc for synthesized compounds.

4.      The designing part should also be revised. What is the reason for replacing the urea linker with an ethylene linker? The author should run some insilico experiments, such as molecular docking study to validate their design strategy.

5.      Authors have provided NMR spectra in supplementary material without showing proton integration in Hydrogen NMR. Please include Hydrogen NMR with proper proton integration.

6.      Authors should also include HPLC purity data of these molecules.

Author Response

Find attached a file with the anwers to reviewer·s comments

Reviewer 2 Report

      In this research, the authors synthesized 14 novel triazole-scaffold derivatives and evaluated their potential usage of them as oncoimmunomodulator agents. In my opinion, the current version of this manuscript fits the scope of International Journal of Molecular Sciences and could be accepted after minor revision.

My specific comments are in detail listed below:

1.     The introduction is poorly written. The authors should discuss the current developments of PD-L1 regulators including inducers and inhibitors carefully and systematically. Besides, the authors should clearly highlight the significance and value of this discovery in the introduction. Some related research or references could be added, such as 10.1038/s41568-021-00431-4, 10.1002/adma.202206121, 10.1038/s41571-022-00601-9, 10.1016/j.jconrel.2022.11.004, 10.1038/S41467-021-25416-7, 10.1016/j.apsb.2022.07.023, 10.1016/j.molcel.2018.07.030, and 10.1038/s41392-020-0200-4.

2.     Figure 2 is of too low quality and should be further modified or improved.

3.     The relationship between c-MYC and PD-L1 expression should be clearly discussed. Some references may be helpful to the authors including 10.1186/s13059-021-02331-0, 10.1186/s13059-021-02331-0, and 10.1002/advs.202002746.

4.     In all the tables, all the data shown are with no decimal place. It’s better to reserve one or two decimal fraction.  

5.     Why these triazole-scaffold derivatives could inhibit PD-L1 should be more clearly discussed.

6.     In Figure 4-5, the error bar should added.

Author Response

Pleas, find attached a file with the answers to reviewer·s comments.

Round 2

Reviewer 1 Report

The authors have revised the ms as suggested by the reviewer.  This ms may be accepted for publication.